# Utility of the Cerebral Organoid Glioma ‘GLICO’ Model for Screening Applications

**DOI:** 10.3390/cells12010153

**Published:** 2022-12-30

**Authors:** Freya R. Weth, Lifeng Peng, Erin Paterson, Swee T. Tan, Clint Gray

**Affiliations:** 1Gillies McIndoe Research Institute, 7 Hospital Road, Wellington 6021, New Zealand; 2Centre for Biodiscovery and School of Biological Sciences, Victoria University of Wellington, Wellington 6021, New Zealand; 3Wellington Regional Plastic, Maxillofacial & Burns Unit, Hutt Hospital, Lower Hutt 5040, New Zealand; 4Department of Surgery, The Royal Melbourne Hospital, The University of Melbourne, Melbourne, VIC 3010, Australia

**Keywords:** glioblastoma, glioblastoma organoids, glioblastoma spheroids, cerebral organoids, glioblastoma stem cells, cancer stem cells, drug screening

## Abstract

Glioblastoma, a grade IV astrocytoma, is regarded as the most aggressive primary brain tumour with an overall median survival of 16.0 months following the standard treatment regimen of surgical resection, followed by radiotherapy and chemotherapy with temozolomide. Despite such intensive treatment, the tumour almost invariably recurs. This poor prognosis has most commonly been attributed to the initiation, propagation, and differentiation of cancer stem cells. Despite the unprecedented advances in biomedical research over the last decade, the current in vitro models are limited at preserving the inter- and intra-tumoural heterogeneity of primary tumours. The ability to understand and manipulate complex cancers such as glioblastoma requires disease models to be clinically and translationally relevant and encompass the cellular heterogeneity of such cancers. Therefore, brain cancer research models need to aim to recapitulate glioblastoma stem cell function, whilst remaining amenable for analysis. Fortunately, the recent development of 3D cultures has overcome some of these challenges, and cerebral organoids are emerging as cutting-edge tools in glioblastoma research. The opportunity to generate cerebral organoids via induced pluripotent stem cells, and to perform co-cultures with patient-derived cancer stem cells (GLICO model), has enabled the analysis of cancer development in a context that better mimics brain tissue architecture. In this article, we review the recent literature on the use of patient-derived glioblastoma organoid models and their applicability for drug screening, as well as provide a potential workflow for screening using the GLICO model. The proposed workflow is practical for use in most laboratories with accessible materials and equipment, a good first pass, and no animal work required. This workflow is also amenable for analysis, with separate measures of invasion, growth, and viability.

## 1. Introduction

Glioblastoma is the most common and most aggressive primary brain tumour, characterised by high recurrence rates and exceptionally poor prognosis [1]. Standard treatment involves intensive multimodal therapy including tumour resection, tumour-treating fields (TTF) radiotherapy or standard radiotherapy, and chemotherapy with Temozolomide (TMZ). The overall median survival is only 16.0 months, with TTF-treated patients reaching an average of 20.9 months [2]. Despite such intensive treatment, less than 30% of patients survive more than 2 years [3]. The inability to effectively treat glioblastoma is due, in part, to the ability of a subpopulation of tumorigenic cells to infiltrate normal brain tissue, preventing complete surgical removal of cancer cells, leading to subsequent recurrence [4]. Improved treatments that target the source of chemotherapy resistance and tumour recurrence are urgently needed, as the overall median survival rates have remained relatively unchanged for 30 years [5].

For glioblastoma to establish beyond the primary site within the brain, this subpopulation of cells must be able to self-renew, generate differentiated tumour cells, and spawn a heterogeneous tumour [6]. Glioblastoma, like hematopoietic malignancies and other solid cancers, has been shown to comprise a small population of cancer stem cells (CSCs) known as glioma stem cells (GSCs) [7], which possess the capacity to recapitulate the heterogeneity of the parent tumour after serial dilution and intracranial implantation into immune-compromised mice [8]. Moreover, GSCs demonstrate particularly infiltrative properties and are thought to be primary contributors to chemotherapy and radiotherapy resistance and tumour recurrence [9]. Therefore, selectively targeting GSC proliferation and invasion in combination with current therapies is seen as a viable option to improve treatment outcomes for glioblastoma patients [10,11].

Proof-of-principle genetic studies have shown that blocking the self-renewal of GSCs leads to prolonged survival in GSC patient-derived mouse efficacy studies [12]. However, identifying optimal GSC-drug targets for clinical translation remains elusive [13]. Establishing disease models that recapitulate the function, heterogeneity, and behaviour of glioblastoma within the brain is therefore a high priority for the evaluation of new therapeutics. Past research into drug treatments for glioblastoma has often relied upon 2D cell culture methods–such as adherent cell culture, in which commonly established cell lines grow in a monolayer attached to a flat surface [14]. Such methods provided a foundation for basic research but have struggled to recapitulate essential features of glioblastoma cells within the brain, such as tumour heterogeneity. Fortunately, recent developments have allowed primary patient-derived tumour cells to be cultured in both 2D and 3D conditions (as both spheroids and organoids). These 3D models can better mimic tumour growth and reflect tumour cell contact within the in vivo tumour microenvironment, features which are not present in a traditional monolayer cell culture context [15]. Additionally, 3D cultures will mimic some of the physical barriers that therapeutic agents encounter when delivered in vivo that are not present in typical 2D cultures, such as hypoxia and impeded diffusion [14,15], ameliorating some of these limitations.

Recent studies have enabled drug screening using patient-derived glioblastoma 3D spheroid cultures [16,17]. Using a 1536-well format and an ultra-high throughput proliferation assay, Quereda et al. [18] demonstrate the applicability of this assay for large-scale high-throughput drug screening. Recent studies have also enabled the generation of a patient-derived glioblastoma-cerebral organoid model in which the resultant tumours phenocopy the patient’s original glioblastoma tumour [19], allowing the study of glioblastoma biology in a human brain model. In this article, we review recent literature on the use of patient-derived glioblastoma organoid models and their applicability for drug screening, as well as providing a potential workflow for screening, using the GLICO model.

## 2. The Necessity of Human Cell-Based Models for Brain Tumour Research

A major challenge in glioblastoma research is to develop disease models that mirror the cellular complexity, aggressive nature and treatment resistance observed in vivo. Previous research models for drug testing and development have relied on an oversimplified approach which has resulted in many of the drugs investigated in traditional preclinical models (2D adherent culture, murine xenografts) yielding suboptimal results in clinical trials, culminating in an FDA approval rate as low as 3% for new oncologic therapies [20].

Established commercial cell lines grown in vitro in 2D adherent culture, offer only modest real-world disease relevance, due to the lack of spatial organisation, cellular heterogeneity, and interaction with non-cancer support cells [21]. Typical established cancer cell lines derived from patient tissues are inefficiently generated and involve extensive adaptation and clonal selection in 2D culture conditions [22]. Only rare clones can be expanded and maintained over many passages; those that can, may have been subject to substantial genetic alterations, and may not recapitulate the genetic complexity of the parent tumour [22]. 2D adherent cultures are not able to accurately translate the function of a human brain and recapitulate the necessary hurdles drugs may encounter in the body–such as hypoxia and impeded diffusion, which calls into question their utility for drug development and testing.

Alternatively, transplantation of patient-derived tumour cells into the brains of immune-compromised mice (xenograft models) provides an in vivo-like model system to investigate the dynamic interplay between GSCs, which drive tumourigenesis, and the non-cancer microenvironment [23]. Unlike invertebrate model systems, tumour development in mice is accompanied by other complex biological processes such as angiogenesis, similar to those in a human cancer [24]. Although there are similarities in the overall cellular architecture between murine and human brains, mice have a much more primitive neocortex which is most highly evolved brain component in humans [25,26]. More importantly, despite similarities in cellular architecture, there are stark differences in gene expression patterns [27], especially of non-neuronal cells such as microglia [28]. Neurons in the human cortex arise from outer radial glia, which are not present (or are very sparse) in rodents [29]. This has important implications, as emerging genomic data from studies in human glioblastoma suggests an important role of non-neuronal cell types such as microglia in the evolution of glioblastoma [30,31]. Further, the genomic evolution of human tumour cells can be altered by the host; brain tumours are not cell autonomous, and they generate properties specific to the host [32]. This may dampen the translational relevance of some aspects of the mouse xenograft model system for drug screening and prediction of human responses in research. Immunodeficient animal husbandry is also a specialised and high-cost procedure with low throughput capabilities for drug screening due to the time taken to generate mice [24].

Recent advances in cell culture technology have led to the development of organoid models that mimic in vivo organ development, increasing our ability to study cellular diversity and complex tissue structures (Figure 1). The establishment of these systems involves either induced pluripotent stem cells (iPSCs) or adult stem cells (AdSC) isolated from a patient that, over extended periods of time, are exposed to various differentiation cues to mimic the developmental process [29]. Stem cells first aggregate to form organ buds and with further long-term culturing, can form organoids [33]. These 3D organoid models are used to study tumour initiation, progression, invasion, and response to drug treatments [34]. Recently, brain organoids developed from human iPSCs have been shown to recapitulate spatial organisation more accurately, cell–cell and cell-niche interactions found in the human brain [35].

## 3. The Role of Glioma Stem Cells in Glioblastoma

The cancer stem cell (CSC) concept of cancer, first proposed more than four decades ago, states that tumour growth is comparable to healthy tissue generation, i.e., it is fuelled by a small population of dedicated stem cells [12]. Decades of developmental and haematopoietic stem cell research [36] support this hypothesis; that cancer growth is maintained exclusively by a small subpopulation of cells with stem cell properties [12,37]. This explains clinical observations such as tumour dormancy, metastasis, and recurrence despite multimodal therapies [38] (Figure 2) and has provided an additional focus into how we should approach cancer treatment [37]. Instead of simply trying to minimise the size of tumours, focus needs to be placed on regulating CSCs–the controllers of long-term growth, tumour recurrence, and invasiveness [39].

In 2003, Singh et al. were the first to isolate CSCs from human brain tumours [40]. These cells exhibit stem cell properties in vitro and express CD133 (CD133+), a transmembrane cell-surface glycoprotein and stem cell biomarker. When transplanted into non-obese diabetic, severe combined immunodeficient mice, these cells are capable of both initiating tumour growth, and recapitulating the original parent tumour in vivo [40]. Glioblastoma CSCs have since been coined GSCs which are quiescent-neoplastic cells [41,42], imbued with multipotency, and self-renewal properties.

Through histological classification, glioblastoma has been traditionally classified as an astrocytoma, although the precise cell type from which the tumour originates is still a controversial issue. Some experts argue that the glioblastoma origin is a subpopulation of neural stem cells, while others propose that it is derived from differentiated astrocytes (Figure 2) [43]. It had long been thought to arise from differentiated glial cells of the central nervous system–hence the name glioblastoma [44]. The eventual isolation of CD133+ (GSC) cells [43] implies that a hierarchy may exist within the tumour cell population [9], because not all cells can maintain the tumour in culture [45]. Normal neural stem cells are also present in the CD133+ population of the foetal brain, suggesting that they may be the cell of origin for glioblastoma [46]. Recently, astrocyte-like neural stem cells in the subventricular zone have been proposed as the cell of origin for glioblastoma [47]. Further research into GSCs and further identification of neural stem cell surface markers may provide insight into this possibility. This could also clarify whether GSCs sit atop a lineage hierarchy or further down as lineage-restricted progenitor cells. Interestingly, emerging evidence suggests that GSCs in glioblastoma are highly heterogenous and rather than occupying an apex, are best described by their cellular state [8]. In support of this, Guilhamon et al. [48] show that chromatin accessibility in GSCs is a critical measure of their invasive property, which is linked to poor survival rates glioblastoma IDH-wild type cohort in the Cancer Genome Atlas, as well as when used in an orthotopic glioblastoma murine model.

GSCs are thought to be the initiators of tumour recurrence and the major contributor to the aggressive nature of glioblastoma [49]. They have been demonstrated to be inherently resistant to conventional therapies through multiple mechanisms, including increased transcription of anti-apoptotic genes and efflux transporters and increased capacity for DNA damage repair [13,24]. GSCs give rise to treatment-resistant clones that aggressively invade normal brain tissue, which is the primary cause of death [38]. Considerable evidence has been generated to support the concept that GSCs are the most biologically and phenotypically relevant cells to the parent tumour in glioblastoma patients [50]. GSCs have the innate capacity for self-renewal and multi-lineage differentiation. These cells are required for tumour initiation, maintenance, and invasion in vivo [51]. Furthermore, compared to other tumour cells, GSCs exhibit increased resistance to chemotherapy and radiotherapy, implicating them in glioblastoma treatment resistance [38] (Figure 3).

Given their vital role in glioblastoma, it seems logical to specifically target GSCs to achieve a durable treatment for glioblastoma. However, therapies aimed at intrinsic mechanisms of GSC proliferation have so far offered only limited success, partly due to the lack of suitable experimental model systems that recapitulate the complex invasive behaviour of GSCs in the human brain. The ability to understand and manipulate complex cancers such as glioblastoma–requires clinically relevant models which encompass the complexity of these tumours and can be used in drug development and testing [52]. Therefore, glioblastoma research models need to aim to recapitulate GSC function within the brain, whilst still being amenable for analysis.

## 4. Patient-Derived Glioblastoma Organoids

Fortunately, the use of patient-derived tumour cells for the development of tumour spheroids and human cerebral organoids to both characterise and model glioblastoma has gone some way to fulfilling this need. Human cerebral organoids are powerful in vitro systems that recapitulate many aspects of human brain development and function [6].

The GSC subpopulation has often been associated with invasion as well as radiotherapy and chemotherapy resistance [38]. The interaction of GSCs with the tumour microenvironment and the ability for quiescence and regeneration is what seems to promote survival and makes these cells difficult to target with chemotherapeutics [32]. GSCs are present throughout the entire glioblastoma tumour. They are localised in both the dense core (hypoxic microenvironment) and at the proliferating edge (increased vascularisation) of the tumour [17,53], surrounded by immune cells such as microglia, which all influence survival and the stem-like state of GSCs [17,30,54].

GSCs isolated from tumour biopsies have been shown to recapitulate the heterogeneity of the patient’s tumour when differentiated in culture or upon xenotransplantation into immune-deficient mice [16,55]. When commercialised glioblastoma cell lines are grown in adherent 2D monolayer cultures without specialised media containing relevant mitogens, they lack the intrinsic heterogeneity and 3D spatial organisation of the patient’s parent tumour [22,56]. Treatment efficacy assays performed on 2D adherent cells often do not translate well to clinical use, with drugs that initially prove effective in the context of 2D cultured cell lines seldom yield equivalent results in a clinical trial setting [14,57,58]. More refined model systems that allow the recapitulation of complex cancer phenotypes and retain the ability to perform detailed analysis are urgently needed.

Patient-derived GSCs can be grown and sustained under specific culture conditions in vitro; with media supplemented with growth factors such as epidermal growth factor and fibroblast growth factor-2 [55]. These conditions can also be used for the expansion of neural stem cells, highlighting the close relationship between these two types of cells [7]. GSCs can be expanded in 2D adherent culture with supplemented media containing relevant GSC growth factors, or as 3D neurospheres [7,59]. Neurospheres, herein referred to as glioblastoma spheroids, can be considered the first “3D model” of glioblastoma, as cells maintain a certain degree of 7 polarisation and 3D spatial organisation [60]. Glioblastoma spheroids, however, have necrotic cores due to the absence of vasculature and can achieve a maximum size of ~300–400 µm before needing disruption and replating to survive [15,61]. On their own, glioblastoma spheroids are unable to form interactions with extracellular matrices or other healthy cells required to generate the specific tumour microenvironment, so are unlikely to completely replicate in vivo GSC behaviour [16].

Formation of human cerebral organoids involves stem cells, either iPSCs or AdSCs, which are sequentially exposed to specific and appropriate exogenous signals to stimulate the developmental process. These conditions allow stem cells to differentiate and aggregate to first form an organ bud/embryoid bodies, and over longer culture, form a cerebral organoid. Mature cerebral organoids can contain differentiated astrocytes, mature neuronal cell types, and even surprisingly microglia-like cells. Ramani et al., surprisingly observed microglial cells and astrocytes in their organoids [62]. Even though it is difficult to explain the development of non-ectodermal related cell types under controlled differentiation conditions, cerebral organoids appear to have some degree of plasticity depending on the differentiation cues.

Due to their potential utility for drug discovery and development, there are many organoid-glioblastoma models that have been developed recently. Hubert et al. [16] cultured minced pieces of resected glioblastoma from patients which successfully formed more complex organoid structures composed of multiple cell types. These organoids recapitulate key glioblastoma features, such as hypoxic gradients, cellular morphology, spacial distribution, and resistance to radiation, however, in vivo GSCs are not autonomous but are heavily influenced by tumour–host cell interactions and the tumour microenvironment [5], which this model does not particularly account for.

Along a similar vein are Bioprinted glioblastoma organoids, which are generated through patient-derived dissociated glioblastoma cells combined with a decellularised porcine brain ‘bioink’ composed of extracellular matrix proteins. On top of this, a layer of human umbilical vein endothelial cells is printed in the same bioink [63]. This model shows the invasion of the organoid into the surrounding endothelial cells and other key features of glioblastoma such as a hypoxic gradient and the presence of multiple cell types. However, there is still a lack of normal brain tissue for interaction as well as the requirement for costly specialised equipment.

Genetically modified cerebral organoids such as neoplastic cerebral organoids (neoCOR) have mutations introduced to induce the expression of oncogenes to cause tumourigenesis within developing iPSC-derived organoids [64,65]. NeoCORs are composed of both healthy and tumour tissue, allowing the study of tumour–brain interactions. However, their ability to recapitulate the heterogeneity of in situ glioblastoma remains to be seen, as they are limited to known mutations of oncogenes that have been studied to date. Such tumour models are advantageous to model glioblastoma initiation, but they hardly recapitulate the genomic complexity of heterogenous patient tumours, so their utility for drug screening remains limited.

A novel approach to overcome this disadvantage has been pioneered by the laboratories of Howard Fine and Amanda Linkous. This approach involves co-culture of patient-derived GSCs in the form of tumour spheroids, with iPSC-derived human cerebral organoids [66]. The authors co-cultured patient-derived GFP-tagged GSC cell lines with mature cerebral organoids and were able to show that GSCs proliferate, form microtubules, and integrate into the organoids. This model has been termed GLIoma Cerebral Organoids (GLICO) [19,67]. The authors showed that different primary patient lines behaved in unique ways, with some showing diffuse invasion, others forming honeycomb-like structures, and some forming regional ‘nodes’ of proliferation [33]. The authors also found that co-cultured GSCs with the greatest invasiveness were more lethal when transplanted into mice [64], suggesting that the observed heterogeneity in growth and invasion in the GLICO model likely reflects certain intrinsic properties of patient-derived GSCs.

The behaviour of cancer cells in this system not only mimic the original tumour, but also maintain key genetic aberrations of the patient’s original tumour [19]. EGFR amplification was identified in two of their primary lines and was maintained in the GLICO model, but was lost in 2D adherent culture [19], indicating that this model may provide a more suitable tumour microenvironment to preserve the genetic characteristics of the tumour in vivo.

Unlike animal brains, human cerebral organoids provide a more species-specific microenvironment which is essential for GSCs to display their inherent characteristics [68]. These models are versatile to characterise various aspects of GSCs–invasion, protrusion, integration, microtubule formation, and interaction with mature neurons of cerebral organoids [69,70]. However, current glioblastoma organoid models suffer from similar weaknesses of most other organoid models, in that they commonly lack vascularisation and innate immune cells [71]. This is important as GSC’s ability to resist many treatment modalities is due in part to interactions with microglial immune cells and increased vasculature within the tumour. Fortunately, methods to produce vascularised and immune-competent organoids are being generated [72,73], with some evidence suggesting microglia innately develop within cerebral organoids generated using the protocol developed by Linkous and Fine [5,74]. Vascularised organoids can be genetically engineered by induced expression of human ETS variant 2 [73]. This allows the cerebral organoids to form a complex vascular-like network. Alternatively, embedding human endothelial cells (hECs) into 9 atrigel before cerebral organoids are incorporated allows the self-assembly of hECs into capillaries at the periphery of organoids, which invade the vascular network [75].

## 5. Prospective Drug Screening Using Patient-Derived Tumourspheres

When glioblastoma spheroids are cultured in basement membrane extracts such as Matrigel or Cultrex, tumour cell invasion can occur and is able to be measured [60]. Spheroid invasion assays are useful for rapid and reproducible assessment of the invasion of tumour cells into a semi-solid medium, making them particularly appropriate for in vitro drug screening for glioblastoma [76]. When supported, the cells can grow outward and invade the matrix in a 3D manner, forming a ‘micro-tumour’ [61]. Cell morphology in these tumourspheres and outgrowths are markedly different from the flat, adherent morphology that cells assume when growing on a solid substrate [77]. Invasion can be easily quantified using an imaging cytometer, an automated read out (Incucyte^®^), or a standard confocal microscope and imaging analysis software (such as TASI) [78]. The significance of this assay, compared to other standard invasion assays, is that tumour cells which invade into the surrounding matrix from the spheroid resemble a “micro-tumour” or a “micro-metastasis”. This, therefore, mimics particularly important aspects of the pathophysiology of a glioblastoma tumour mass, such as an interconnected network of tumour microtubes, which aid in the invasion of normal host tissue [79]. Additionally, cells within glioblastoma spheroids may experience hypoxia and nutrient deprivation which, through changes in gene expression, can promote migration and invasion [80]. This is something that cannot be achieved with 2D assays.

Moreover, all of these techniques are starting to become more accessible, in high-throughput formats, using specific 96- and 384-well plates [18] and the latest imaging and bioinformatic technology [81,82], allowing more complex 3D assays to be used in drug screening. A limitation of the invasion assay method as for any such assays is the difficulty in distinguishing between invasion and proliferation, which the tumour cells are likely to undergo during the assay time frame [79]. For this reason, a parallel 3D growth assay may be performed to evaluate specific effects of any inhibitory or stimulatory agents [76]. If careful dose–response studies are performed, it may be possible to select concentrations of the desired drug/s that minimises the effects on proliferation to assess the effects of the drug on invasion only. For example, Vinci, Box and Eccles [79] have shown that the HSP90 inhibitor 17-AAG inhibits U-87 MG 3D tumour spheroid invasion at 24 h and at concentrations below the concentration inhibiting 3D growth by 50% (GI50). A prospective drug screen using patient-derived tumour spheres can be performed to ameliorate two of the main limitations relating to the generation of cerebral organoid models: time taken to culture and cost of materials.

## 6. Drug Screening Using a Patient-Derived Glioblastoma-Cerebral Organoid Model

Taking advantage of the GLICO model for higher-scale compound screening has not yet been achieved [83], likely due to the prohibitive costs and time taken to generate both human cerebral organoids, and a high-throughput drug screening system suitable for these 3D cultures. Still, progress has been made towards proving the useability of the GLICO model for these applications.

Utilising a genetically engineered luciferase-based assay to measure proliferation, Linkous et al. [19] were able to show that different GLICO models utilising different patient-derived tumours have differential sensitivities to chemotherapy (TMZ, BCNU) and radiotherapy. Interestingly, when the same patient samples were cultured in 2D, they showed equal susceptibility to TMZ and BCNU which was not the case for the GLICO model where differential sensitivity was seen [19]. The use of secreted luciferase as an indirect measure for determining viable glioblastoma cells within organoids offers simplicity, sensitivity, and scalability, making it amenable for high-throughput drug screening [84]. The limitation of the luciferase system, and other cell viability assays, is the inability to address key complex features of glioblastoma such as invasion and cell morphology [83]. Therefore, we propose live-cell immunofluorescent imaging of GSC invasion into GLICOs alongside viability measures (Figure 3). Our proposed workflow (Figure 4), for smaller scale screening using the GLICO model is particularly practical for use in most laboratories with accessible materials, accessible equipment, good for first pass, and no animal work required. This model is also amenable for analysis, with separate measures of invasion, growth, and viability. This workflow considers particularly important aspects of GSC function and how they may be altered upon therapeutic treatments. The reduction in cost and time taken to culture arises from the use of spheroids for preliminary screening prior to use of the GLICO model. The cost of generating and screening spheroids compared to organoids is significantly lower when using spheres which have specific and defined methods for quantification.

## 7. Conclusions and Perspectives

The ability to understand and manipulate cancers such as glioblastoma requires the disease models to be clinically and translationally relevant, replicating the cellular heterogeneity of such cancers. Therefore, brain cancer research models need to aim to recapitulate GSC function, whilst remaining amenable for analysis. Our proposed workflow of preliminary drug screening using glioblastoma spheroids as part of the GLICO model utilises patient-derived glioblastoma cells and cerebral organoids, providing the unique capability to investigate tumour–brain interactions. Despite their usefulness, the versatility and reliability of brain organoids in modelling glioblastoma remains to be standardised, with cost and time to culture being some of the greatest limitations. The GLICO model suffers from similar weaknesses as other organoid models, with the absence of vasculature and immune cells, which are necessary for accurate recapitulation of the tumour microenvironment. However, recent research is making significant steps to ameliorate these limitations. Our proposed model for smaller-scale drug screening is particularly practical for use in most laboratories, or for those wanting to move into 3D cell culture/organoid research; with accessible materials, and accessible equipment, good for the first pass, and no animal work required.

## Figures and Tables

**Figure 1 cells-12-00153-f001:**
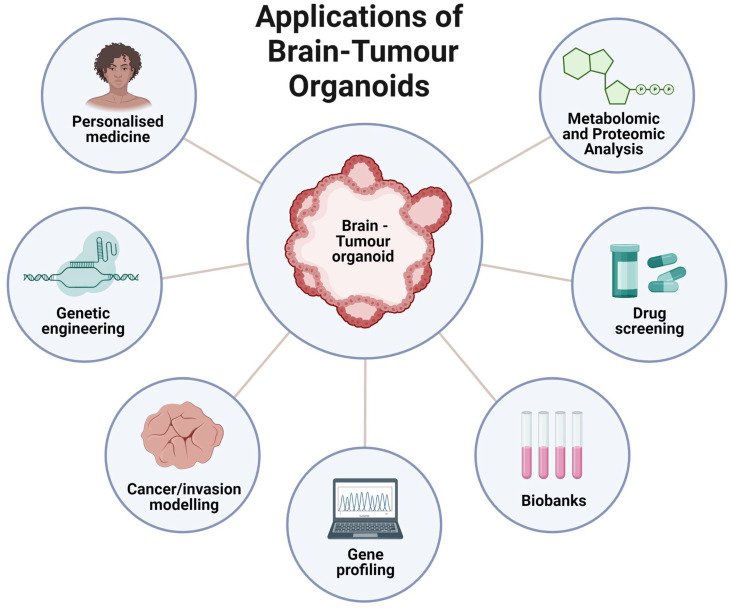
Applications of Brain-Tumour Organoids. Recent advances in cell culture technology have allowed the use of organoids as model systems for a myriad of applications such as personalised medicine, genetic engineering, cancer and invasion modelling, gene profiling, primary cell/tumour biobanks, drug screening, and metabolomic/ proteomic analyses. Created with Biorender.com accessed on 25 September 2022.

**Figure 2 cells-12-00153-f002:**
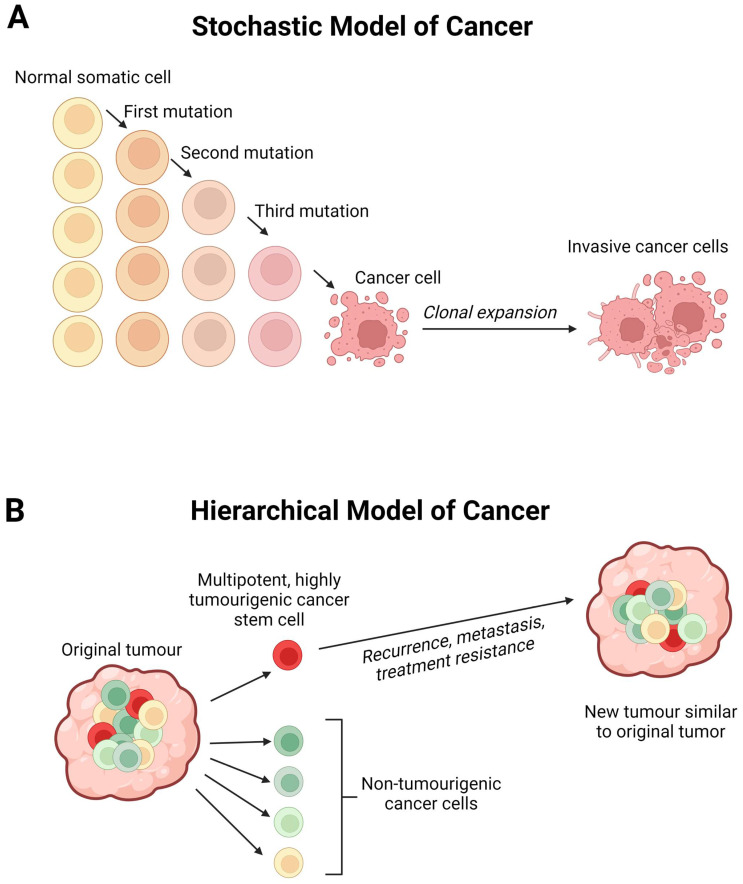
(**A**) The stochastic model of cancer proposes that a normal somatic cell accumulates oncogenic mutations in a stepwise manner and becomes a cancer cell that undergoes clonal expansion to form a tumour. (**B**) The hierarchical model of cancer proposes the presence of a highly tumourigenic cancer stem cell (CSC) sitting atop the tumour cellular hierarchy and divides asymmetrically to form non-tumourigenic cancer cells that form the bulk of the tumour, and identical CSCs that form new tumours like the original tumour. Created with Biorender.com accessed on 1 October 2022.

**Figure 3 cells-12-00153-f003:**
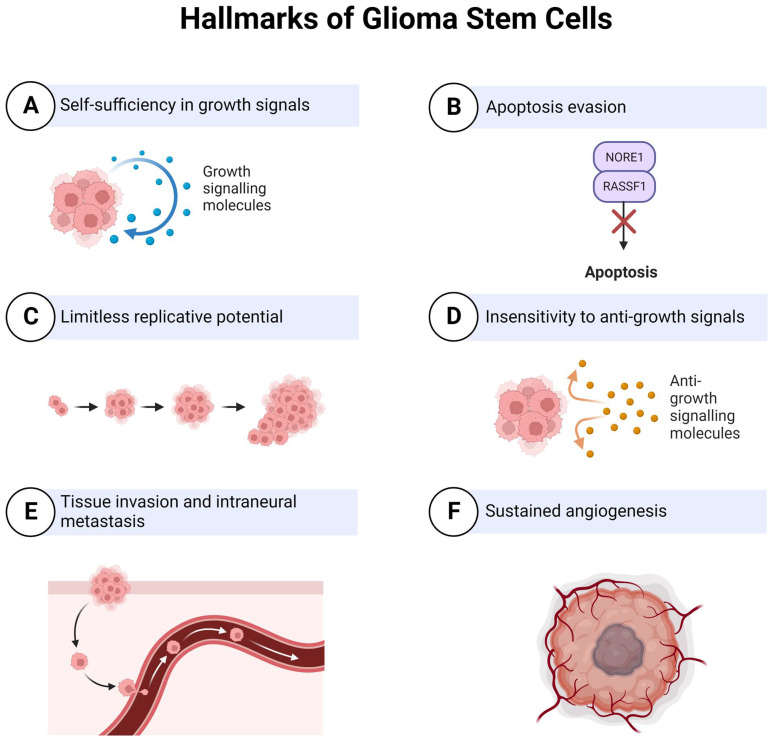
Hallmarks of Glioma Stem Cells (GSCs). GSCs are the primary contributors to the aggressive nature of glioblastoma. The features that make these cells particularly invasive are self-sustained growth signalling, evasion of programmed cell death, limitless replicative potential with minimal cell senescence, tissue invasion and intraneural metastasis and sustained angiogenesis/increased vascularisation. Adapted from Biorender.com accessed on 1 October 2022.

**Figure 4 cells-12-00153-f004:**
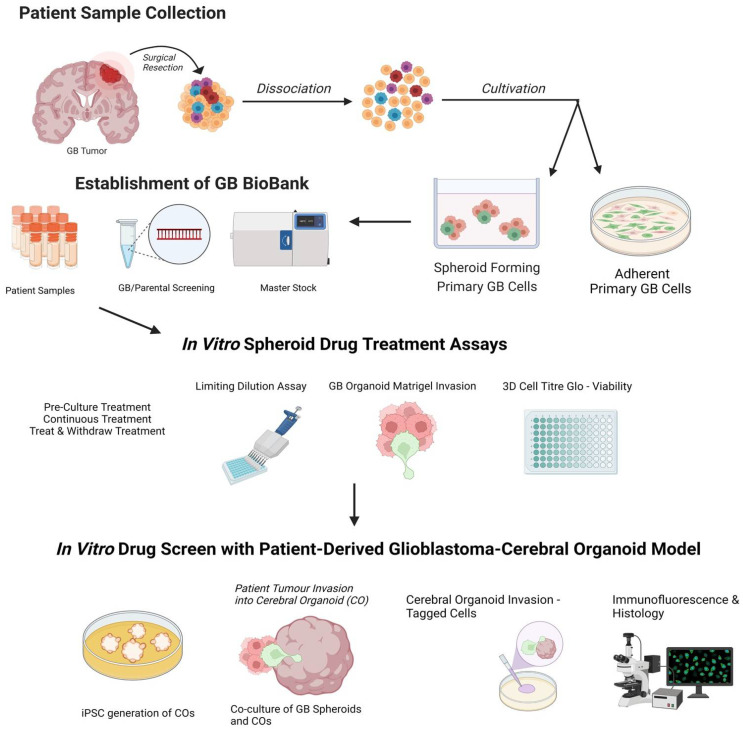
A potential workflow for drug screening using the GLICO model: 1. Generation of spheroid forming primary glioblastoma (GB) cells and biobanking of patient samples. 2. Prospective drug screen on glioblastoma spheroids. 3. Refined drug screen using patient-derived glioblastoma-cerebral organoid (GLICO) model. Created with Biorender.com accessed on 25 September 2022.

## Data Availability

Not applicable.

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
