# Peer review of "Utility of the Cerebral Organoid Glioma ‘GLICO’ Model for Screening Applications"

_cells, 2022, doi:10.3390/cells12010153_

Round 1

Reviewer 1 Report

Weth et al. provided a well-written and timely review on a topic that is becoming increasingly relevant in the GBM field. Recent landmark papers are shedding light on how the brain cells and the brain microenvironment is affecting GBM pathogenesis and treatment outcomes. Therefore, developing and understanding how the GBM tumor cells interact with human brain cells and their microenvironments in an in vitro system is becoming paramount. Overall, the authors introduced the topic well but were light on areas that should have defined this review – the GLICO model itself. I suggest focusing on this topic by adding more section(s) on GLICO, as it currently reads more like a general overview of the cell culture models in GBM. I have suggested some ideas that could help strengthen this manuscript and become a valuable resource to the GBM field. 

Suggestions

The median survival is currently close to 16 months, with TTF-treated patients reaching an average of 20 months. Please update references to reflect this.

The authors might want to update their reference for therapy resistance on page 2 of the introduction. They can keep references no. 9 and 10, but might want to consider the following review. Alves, A.L.V., Gomes, I.N.F., Carloni, A.C. et al. Role of glioblastoma stem cells in cancer therapeutic resistance: a perspective on antineoplastic agents from natural sources and chemical derivatives. Stem Cell Res Ther 12, 206 (2021).

Reference 29 on page 3 describes GBM organoids, not brain organoids, yet the authors have cited 29 to back up their brain organoid statement. Please change.

Reference 33 on line 133 is quite outdated. Please update with references from the last 5 years in glioma cancer stem cells.

The authors should discuss the neuron-GBM and the astrocyte-GBM cell communication axes. These are now well-published phenomena, and it is highly relevant to the GLICO model. Do the neurons and astrocytes in the GLICO model form synaptic connections with the tumour cells?

The review is titled “Utility of the cerebral organoid glioma model (GLICO)”. Yet, they focus quite heavily on glioma stem cells (GSCs) and less on how GSCs are used in generating a cerebral organoid model. For the purposes of this review, I would have liked a section before section 5 (so, the current section 5 will become section 6) that reviews how the GLICO model was made, what versions of the models exist, how they were characterised, and what developments are on the horizon to mitigate the limitations of this model (see below).  

The authors stated that GSC’s ability to resist many treatment modalities was due to being in a hypoxic microenvironment that has increased vascularisation and is surrounded by microglia (lines 174-179). Yet, classical brain organoids lack vasculature and microglia (which the authors briefly mention in line 244). This would be further improved if the authors could discuss how these limitations could be overcome. Also, recent developments have allowed both vasculature and microglia to be incorporated into a brain organoid. 

Vasculature = Cakir, B., Xiang, Y., Tanaka, Y. et al. Engineering of human brain organoids with a functional vascular-like system. Nat Methods 16, 1169–1175 (2019).  Microglia = Zhang, W., Jiang, J., Xu, Z. et al. Microglia-containing human brain organoids for the study of brain development and pathology. Mol Psychiatry (2022). 

I think it would be fitting to discuss GLICO in light of these developments to truly recapitulate the tumor microenvironment.

Author Response

Weth et al. provided a well-written and timely review on a topic that is becoming increasingly relevant in the GBM field. Recent landmark papers are shedding light on how the brain cells and the brain microenvironment is affecting GBM pathogenesis and treatment outcomes. Therefore, developing and understanding how the GBM tumor cells interact with human brain cells and their microenvironments in an in vitro system is becoming paramount. Overall, the authors introduced the topic well but were light on areas that should have defined this review – the GLICO model itself. I suggest focusing on this topic by adding more section(s) on GLICO, as it currently reads more like a general overview of the cell culture models in GBM. I have suggested some ideas that could help strengthen this manuscript and become a valuable resource to the GBM field. 

The median survival is currently close to 16 months, with TTF-treated patients reaching an average of 20 months. Please update references to reflect this.

We agree with the reviewer and have changed text and included appropriate references to reflect these changes, see lines 13, 37,38.

  1. Stupp et al., “Effect of Tumor-Treating Fields Plus Maintenance Temozolomide vs Maintenance Temozolomide Alone on Survival in Patients With Glioblastoma,” JAMA, vol. 318, no. 23, pp. 2306–2316, Dec. 2017, doi: 10.1001/jama.2017.18718.

The authors might want to update their reference for therapy resistance on page 2 of the introduction. They can keep references no. 9 and 10, but might want to consider the following review. Alves, A.L.V., Gomes, I.N.F., Carloni, A.C. et al. Role of glioblastoma stem cells in cancer therapeutic resistance: a perspective on antineoplastic agents from natural sources and chemical derivatives. Stem Cell Res Ther 12, 206 (2021).

Authors thank the reviewer for highlighting an excellent review. We have changed text and references have been included, lines 69.

  1. L. V. Alves et al., “Role of glioblastoma stem cells in cancer therapeutic resistance: a perspective on antineo-plastic agents from natural sources and chemical derivatives,” Stem Cell Res Ther, vol. 12, p. 206, Mar. 2021, doi: 10.1186/s13287-021-02231-x.

Reference 29 on page 3 describes GBM organoids, not brain organoids, yet the authors have cited 29 to back up their brain organoid statement. Please change.

References updated, see line 143.

  1. S. Agboola, X. Hu, Z. Shan, Y. Wu, and L. Lei, “Brain organoid: a 3D technology for investigating cellular com-position and interactions in human neurological development and disease models in vitro,” Stem Cell Research & Therapy, vol. 12, no. 1, p. 430, Jul. 2021, doi: 10.1186/s13287-021-02369-8.

Reference 33 on line 133 is quite outdated. Please update with references from the last 5 years in glioma cancer stem cells.

Reference updated, see line 157

  1. Marzagalli, F. Fontana, M. Raimondi, and P. Limonta, “Cancer Stem Cells—Key Players in Tumor Relapse,” Cancers, vol. 13, no. 3, Art. no. 3, Jan. 2021, doi: 10.3390/cancers13030376.

The authors should discuss the neuron-GBM and the astrocyte-GBM cell communication axes. These are now well-published phenomena, and it is highly relevant to the GLICO model. Do the neurons and astrocytes in the GLICO model form synaptic connections with the tumour cells?

Agreed. The authors have included text to discuss neuron-GBM and the astrocyte-GBM cell communication axes. Lines 365 to 390.

The review is titled “Utility of the cerebral organoid glioma model (GLICO)”. Yet, they focus quite heavily on glioma stem cells (GSCs) and less on how GSCs are used in generating a cerebral organoid model. For the purposes of this review, I would have liked a section before section 5 (so, the current section 5 will become section 6) that reviews how the GLICO model was made, what versions of the models exist, how they were characterised, and what developments are on the horizon to mitigate the limitations of this model (see below).  

Agreed. We have included section 5 which describes human glioblastoma cerebral organoids.

Lines 288 to 386

The authors stated that GSC’s ability to resist many treatment modalities was due to being in a hypoxic microenvironment that has increased vascularisation and is surrounded by microglia (lines 174-179). Yet, classical brain organoids lack vasculature and microglia (which the authors briefly mention in line 244). This would be further improved if the authors could discuss how these limitations could be overcome. Also, recent developments have allowed both vasculature and microglia to be incorporated into a brain organoid. 

Vasculature = Cakir, B., Xiang, Y., Tanaka, Y. et al. Engineering of human brain organoids with a functional vascular-like system. Nat Methods 16, 1169–1175 (2019).  Microglia = Zhang, W., Jiang, J., Xu, Z. et al. Microglia-containing human brain organoids for the study of brain development and pathology. Mol Psychiatry (2022). 

I think it would be fitting to discuss GLICO considering these developments to truly recapitulate the tumor microenvironment.

Agreed. We have included these changes in section 5 outlined above and the new references suggested by the reviewer has been incluced. Lines 285 to 300

Reviewer 2 Report

The manuscript reviews the literature on glioblastoma cerebral organoids and their usefulness as a model for screening applications, although it is not fully clear whether the intention is a literature review or a perspective article and this should be made clearer. The text is written well and there is a logical flow, but in its present form the text contains several conceptual errors. The review is timely and relevant for the field because organoid models are gaining increasing popularity as models for glioblastoma research. The authors should make an effort to include more references of original work instead of review articles. The manuscript would benefit from a separate section on brain organoids – these are an essential component of the GliCO model but are only mentioned briefly at the end of the review.  Because the GliCO model is not widely used at the moment, the authors should include a section to discuss other glioblastoma organoid models and explain the differences of the GliCO model. While well written and in good English, there are a number of typos in the manuscript which should be carefully edited to remove these.

Specific comments:

line 53 – 68: This paragraph should mention that modelling tumor heterogeneity requires primary patient-derived lines rather than established cell lines. Most 2D culture models use established lines. Also, authors should mention that neurospheres or tumor spheres are also 3D culture models and better discuss the advantages of organoids over sphere models.

line 80/81: I would recommend rephrasing ‘Previous models have relied on a reductionist approach’ – while organoid models are arguably superior to monolayers, they are still a reductionist model.

line 88-101: this paragraph needs some editing for clarity. The authors make a case that non-neuronal cells in the human brain have different gene expression patterns to mice (l.92). Then they follow this up with a statement about the developmental origin of human cortical neurons. The they mention that microglia have an important role in human glioblastoma. Is there evidence showing that murine microglia in GBM xenograft models have different gene expression patterns compared to human tumors? This should be referenced here. Similarly, any evidence that the developmentally different origins of human cortical neurons impact GBM development and progression should be included here. In the final sentence of this paragraph (‘calling into question the translational relevance of some aspects…’), details should be added which aspects the authors mean, supported by references. It would be helpful to spell out where xenograft models are possibly useful in research, and where these models are inadequate.

line 121-123: It would be helpful for non-specialist readers to expand how the CSC hypothesis (potentially) explains tumor recurrence, dormancy and metastasis.

line 131-132: This sentence is confusing and contains statements that are borderline wrong. GSCs are not automatically quiescent and there are many studies in the field that link cancer stem cells and proliferation. GSCs are also not pluripotent (capable of differentiating into all three germ layers).

line 133-147: This paragraph should be edited for further clarity. Firstly, the current top candidate for cell of origin (neural stem cells) is a type of glial cell. Secondly, this paragraph should discuss and reference recent works investigating the origins of GBM and GSCs, e.g. from the Parada lab, but also e.g. Lee et al. Nature 2018 560(7717):243-247. The citation (9) in line 144 is incorrectly referenced.

line 154-155: I would argue that multi-lineage differentiation is not a defining criterion of GSCs (e.g., see Lathia et al. Genes Dev 2015 29(12):1203-17). The main definition of a GSC is its ability to initiate tumor growth.  

The reference to metastasis should be removed from Figure 2 as GBM and gliomas rarely metastasize outside the CNS.

line 177-178: The statement ‘hypoxia and increased vascularisation’ should be edited for clarity, otherwise it might seem paradox.

line 182-183: The sentence ‘When GSCs are grown in adherent 2D monolayer cultures…’ should be tempered, as GSCs can be cultured on laminin in the presence of mitogens and retain their properties (e.g. see Pollard et al. Cell Stem Cell 2009 4(6):568-80). The authors reference this further below, but I think this paragraph needs some clarification.

line 225-232: I fail to follow why it is desirable to identify chemotherapeutics that block invasion with minimised effects on proliferation. Surely targeting both invasion and proliferation should be a beneficial strategy for anti-cancer therapy? This point needs to be explained in more detail. Also, because the whole section is on GSCs it would be great if the authors could find an example that does not rely on U-87MG and instead uses primary patient derived GSC sphere models.

line 263-277: The authors review work from the Fine lab which compared results from GliCO models to 2D adherent cultures, but a more suitable comparison would be to compare GliCOs to other 3D culture models. The authors should consider including this in their discussion. 

line 283-284 and Figure 3: The authors refer to their ‘proposed model for smaller scale screening’, but this is not really explained in the text. It is unclear how this model would overcome the current problems of ‘extensive costs and time taken to generate’ organoid models (line 269/270) and this should be clarified. The legend of Figure 3 should contain more details to explain the figure content better. There is reference to points 1-3 in the figure legend, which are missing in the figure. 

line 294-307: the text in this section is largely redundant to the previous section and should be revised to provide a more compelling conclusion. It is unclear in this section how GliCOs better recapitulate GSC function over e.g., sphere models and this should be explained in more detail. From my understanding the advantage of GliCOs is that they contain several aspects of the brain environment which sphere models lack, but it is not clear whether GliCOs preserve GSC function and/or heterogeneity more than sphere models.

Author Response

Reviewer #4

The manuscript reviews the literature on glioblastoma cerebral organoids and their usefulness as a model for screening applications, although it is not fully clear whether the intention is a literature review or a perspective article and this should be made clearer. The text is written well and there is a logical flow, but in its present form the text contains several conceptual errors. The review is timely and relevant for the field because organoid models are gaining increasing popularity as models for glioblastoma research. The authors should make an effort to include more references of original work instead of review articles. The manuscript would benefit from a separate section on brain organoids – these are an essential component of the GliCO model but are only mentioned briefly at the end of the review.  Because the GliCO model is not widely used at the moment, the authors should include a section to discuss other glioblastoma organoid models and explain the differences of the GliCO model. While well written and in good English, there are several typos in the manuscript which should be carefully edited to remove these.

The authors would like to thank reviewer #4 for their constructive feedback. We have incorporated

 The suggested changes and included up to date references where necessary and feel the review has benefited from these amendments.

Specific comments:

line 53 – 68: This paragraph should mention that modelling tumor heterogeneity requires primary patient-derived lines rather than established cell lines. Most 2D culture models use established lines.

We would agree and have included text to add clarity to the original text, see lines 70 to 86.

Also, authors should mention that neurospheres or tumor spheres are also 3D culture models and better discuss the advantages of organoids over sphere models.

We agree. The discussion of neurosphere/tumour spheres (spheroids) has been moved and included better discussed in section 5

line 80/81: I would recommend rephrasing ‘Previous models have relied on a reductionist approach’ – while organoid models are arguably superior to monolayers, they are still a reductionist model.

The authors have rephrased the term. See lines 98 to 100.

line 88-101: this paragraph needs some editing for clarity. The authors make a case that non-neuronal cells in the human brain have different gene expression patterns to mice (l.92). Then they follow this up with a statement about the developmental origin of human cortical neurons. The they mention that microglia have an important role in human glioblastoma.

The authors have reworded and restructured and included more relevant references in this section. Please see below. See lines 122-290

Is there evidence showing that murine microglia in GBM xenograft models have different gene expression patterns compared to human tumors? This should be referenced here.

Agreed. We have altered text and included a relevant reference.

Similarly, any evidence that the developmentally different origins of human cortical neurons impact GBM development and progression should be included here.

Tumour-host interaction discussion section has been included, reworded and restructured paragraph to reflect the reviewers comment.

In the final sentence of this paragraph (‘calling into question the translational relevance of some aspects…’), details should be added which aspects the authors mean, supported by references.

It would be helpful to spell out where xenograft models are possibly useful in research, and where these models are inadequate.

The authors have rephrased and removed some of discussion related to and included more reference to the utility of the xenograph animal model in drug screening studies. See lines 122-190.

A larger more drawn-out section with regards to xenograph model in our opinion is of limited use to the scope of the current review. We acknowledge it is a robust model of GBM with many useful clinical applications however, the use of xenograph models/animals for higher throughput drug screening compared to prospective organoids is minimal …. ? However, to acknowledge their utility for other applications, we have amended text and included text to highlight the potential of the GBM xenograph model.

line 121-123: It would be helpful for non-specialist readers to expand how the CSC hypothesis (potentially) explains tumour recurrence, dormancy, and metastasis.

We have altered existing text and added a further figure to explain to the non-specialist reader these points.

line 131-132: This sentence is confusing and contains statements that are borderline wrong. GSCs are not automatically quiescent and there are many studies in the field that link cancer stem cells and proliferation. GSCs are also not pluripotent (capable of differentiating into all three germ layers).

The authors would like to highlight that the original text did not state that GSC’s were or were not ‘automatically’ quiescent and acknowledge they proliferate. We simply state that they are ‘quiescent neoplastic cells’ as when not proliferating, they are in a state of quiescence. The statement we were trying to articulate, is that quiescence is a reversible state or period of dormancy in these neoplastic cells – as it is also an important and unique property of stem cells. However, we understand the wording may have caused some confusion. Therefore, we have altered in text and included below references for quiescent cancer cells. See lines 177-178.

  1. N. Nik Nabil et al., “Towards a Framework for Better Understanding of Quiescent Cancer Cells,” Cells, vol. 10, no. 3, p. 562, Mar. 2021, doi: 10.3390/cells10030562.

  1. Chen, J. Dong, J. Haiech, M.-C. Kilhoffer, and M. Zeniou, “Cancer Stem Cell Quiescence and Plasticity as Major Challenges in Cancer Therapy,” Stem Cells Int, vol. 2016, p. 1740936, 2016, doi: 10.1155/2016/1740936.

line 133-147: This paragraph should be edited for further clarity. Firstly, the current top candidate for cell of origin (neural stem cells) is a type of glial cell. Secondly, this paragraph should discuss and reference recent works investigating the origins of GBM and GSCs, e.g. from the Parada lab, but also e.g. Lee et al. Nature 2018 560(7717):243-247. The citation (9) in line 144 is incorrectly referenced.

Agreed. The authors have amended the text and included the recent works/references from Parada lab. See lines 181 to 188.

line 154-155: I would argue that multi-lineage differentiation is not a defining criterion of GSCs (e.g., see Lathia et al. Genes Dev 2015 29(12):1203-17). The main definition of a GSC is its ability to initiate tumor growth.

We have included this suggestion in text. Discussion of GSC ability to initiate and maintain tumour growth followed that sentence but authors have rearranged for clarity.

The reference to metastasis should be removed from Figure 2 as GBM and gliomas rarely metastasize outside the CNS.

We agree, they do not metastasis outside the CNS and have clarified this by altering figure/legend text to include ‘intraneural metastasis’.

line 177-178: The statement ‘hypoxia and increased vascularisation’ should be edited for clarity, otherwise it might seem paradox.

The authors have rephrased the statement to remove potential paradoxical meaning. See lines 232 to 234.

line 182-183: The sentence ‘When GSCs are grown in adherent 2D monolayer cultures…’ should be tempered, as GSCs can be cultured on laminin in the presence of mitogens and retain their properties (e.g. see Pollard et al. Cell Stem Cell 2009 4(6):568-80). The authors reference this further below, but I think this paragraph needs some clarification.

The authors have rephrased the statement to aid clarity. See lines 237 to 239.

line 225-232: I fail to follow why it is desirable to identify chemotherapeutics that block invasion with minimised effects on proliferation. Surely targeting both invasion and proliferation should be a beneficial strategy for anti-cancer therapy? This point needs to be explained in more detail. Also, because the whole section is on GSCs it would be great if the authors could find an example that does not rely on U-87MG and instead uses primary patient derived GSC sphere models.

The authors would like to highlight, the section is regarding screening of drugs on spheroids and not clinical outcome. We are not suggesting that it is desirable to block invasion and not proliferation. We agree with the reviewer in that targeting both is obviously the most prudent course for beneficial outcomes. However, we are simply stating that if one wanted to assess the specific effects of drugs on invasion only, or proliferation only. There are studies that have shown this. We agree with the reviewer that these are of little use in clinical treatment setting, but can be useful in elucidating the molecular drivers and mechanisms that govern or are involved in either invasion or proliferation separately. To aid clarity we have altered text to read more clearly.

line 263-277: The authors review work from the Fine lab which compared results from GliCO models to 2D adherent cultures, but a more suitable comparison would be to compare GliCOs to other 3D culture models. The authors should consider including this in their discussion. 

Agreed. We have included section 5 which describes human glioblastoma cerebral organoids and compares GLICO model vs other 3D culture models. Lines 288 to 386.

line 283-284 and Figure 3: The authors refer to their ‘proposed model for smaller scale screening’, but this is not really explained in the text. It is unclear how this model would overcome the current problems of ‘extensive costs and time taken to generate’ organoid models (line 269/270) and this should be clarified.

The reduction in cost arises from the use of spheroids for preliminary screening prior to use of spheroids mixed with organoids (GLICO model). Therefore, the costs of generating spheroids compared to screening everything on the GLICO model are substantially lower when using spheroids.

We have altered the text to be clearer and made clearer reference to the figure which explains our workflow/model.

The legend of Figure 3 should contain more details to explain the figure content better. There is reference to points 1-3 in the figure legend, which are missing in the figure. 

Agreed, the figure and figure legend has been updated.

line 294-307: the text in this section is largely redundant to the previous section and should be revised to provide a more compelling conclusion. It is unclear in this section how GliCOs better recapitulate GSC function over e.g., sphere models and this should be explained in more detail. From my understanding the advantage of GliCOs is that they contain several aspects of the brain environment which sphere models lack, but it is not clear whether GliCOs preserve GSC function and/or heterogeneity more than sphere models.

Agreed. We amended the text to provide a compelling conclusion and we have also included a lot of discussion surrounding recapitulation of parent tumour phenotype vs model in section 5.

Again, the authors would like to thank all of the reviewers for their constructive feedback. We have endeavoured to incorporate the suggested changes and included up to date references where necessary and feel the review has greatly benefited from these amendments.

Kind regards

Freya Weth and Dr Clint Gray

Reviewer 3 Report

Thank you for the opportunity to review the paper. First of all - I did not check the manuscript for plagiarism - but this is a Yes/No answer - I couldn't tell.

The authors describe their workflow and the translational background including the necessity to provide drug screening for personalized medicine in glioma.

Overall, the paper is well written and the methodology is sound. It adds value to the community by its stepwise and educational approach to establishing a patient derived stem cell database for further research and treatment screening.

I recommend the paper for publication.

Author Response

We would like to thank the reviewer for their kind comments and review of our manuscript.

Reviewer 4 Report

The work is well-presented and shows all the aspects relevant to the field.

One minor point: line 33 one should not write GBM is ..primary brain cancer, but instead brain tumor

Author Response

One minor point: line 33 one should not write GBM is ..primary brain cancer, but instead brain tumor

Agreed and amended, see line 34